# An Epidemiological Update on *Anisakis* Nematode Larvae in Red Mullet (*Mullus barbatus*) from the Ligurian Sea

**DOI:** 10.3390/pathogens12111366

**Published:** 2023-11-18

**Authors:** Dáša Schleicherová, Vasco Menconi, Barbara Moroni, Paolo Pastorino, Giuseppe Esposito, Serena Canola, Marzia Righetti, Alessandro Dondo, Marino Prearo

**Affiliations:** 1Istituto Zooprofilattico Sperimentale del Piemonte, Liguria e Valle d’Aosta, via Bologna 148, 10154 Torino, Italy; dasa.schleicherova@izsto.it (D.S.); vmenconi@izsvenezie.it (V.M.); paolo.pastorino@izsto.it (P.P.); giuseppe.esposito@izsto.it (G.E.); serena.canola@gmail.com (S.C.); marzia.righetti@gmail.com (M.R.); alessandro.dondo@izsto.it (A.D.); marino.prearo@izsto.it (M.P.); 2Department of Life Sciences and Systems Biology (DBIOS), University of Turin, via Accademia Albertina 13, 10123 Turin, Italy

**Keywords:** *Mullus barbatus*, *Anisakis pegreffii*, *Hysterothylacium* spp., Mediterranean Sea, marine biodiversity, global warming, fish-borne parasites, zoonoses

## Abstract

Red mullet (*Mullus barbatus*) is a commercially relevant fish species, yet epidemiological data on anisakid nematode infestation in *M. barbatus* are scarce. To fill this gap, we report the occurrence of *Anisakis* larvae in red mullet in the Ligurian Sea (western Mediterranean). This survey was performed between 2018 and 2020 on fresh specimens of *M. barbatus* (n = 838) from two commercial fishing areas (Imperia, n = 190; Savona, n = 648) in the Ligurian Sea. Larvae morphologically identified as *Anisakis* spp. (n = 544) were characterized using PCR-RFLP as *Anisakis pegreffii*. The overall prevalence of *A. pegreffii* was 24.46%; the prevalence at each sampling site was 6.32% for Imperia and 29.78% for Savona. Furthermore, 3300 larvae of *Hysterothylacium* spp. were detected in the visceral organs of fish coinfected with *A. pegreffii*, showing that coinfection with two parasitic species is not rare. This study provides a timely update on the prevalence of ascaridoid nematodes in red mullet of the Ligurian Sea, an important commercial fishing area in the Mediterranean.

## 1. Introduction

Two fish species of the Mullidae family populate the Mediterranean Sea: the striped mullet or surmullet (*Mullus surmuletus*) and the red mullet (*Mullus barbatus*).

Red mullet is a benthic fish inhabiting muddy or sandy seabeds, with a predilection for a depth between 10 and 300 m [1], while surmullet is more typically found in shallow waters. Both species feed on benthic invertebrates such as polychaetes, crustaceans, and mollusks [2]. These fish are known for their delicate flavor and have been considered an important source of food in Italy since ancient times, being included in many regional traditional recipes.

Red mullet is a valued commercial species in the Mediterranean, where it is usually caught through bottom trawling and small-scale fisheries [3]. In addition to being widespread in the Mediterranean, it is also found in the Black Sea and along the Atlantic coast of Europe and Africa [4,5,6]. The annual catch of red mullet by commercial fisheries in Italy is 6500 tons [7], constituting a major share of the fishery sector in this country.

A serious public health threat associated with the consumption of raw or undercooked red mullet meat is anisakiasis, a fish-borne zoonotic parasitosis that causes economic losses worldwide [8]. Fish infected with third-stage larvae (L3) of the genus *Anisakis*, including *A. simplex* sensu stricto (s.s.) and *A. pegreffii* [9], are considered the primary cause of human anisakiasis worldwide, although it is unclear whether other species of the genus *anisakis* have zoonotic potential or not [10]. The life cycle of anisakid nematodes involves at least two different hosts, with marine mammals as final hosts, where adult nematodes develop in the stomach. Crustaceans (i.e., euphasiids) act as intermediate hosts, where free-swimming L3 larvae are ingested and develop in the hemocoel, and many different fish families and cephalopods function as intermediate or paratenic hosts [11,12]. In this complex scenario, humans play a marginal role as dead-end hosts, in which *Anisakis* larval stages do not develop into adults, though they can still cause mild to severe gastrointestinal illness [13,14]. The typical clinical manifestations of anisakiasis include acute abdominal pain, occasionally accompanied by nausea, vomiting, and fever within 12 h after consumption of infected fish [15,16]. Allergic reactions (e.g., angioedema, urticaria, anaphylaxis) have also been reported [17,18].

Epidemiological data on this food-borne parasitic nematode in fish species such as the red mullet are still scant or outdated in the Mediterranean Sea. Manfredi et al. [19] reported a prevalence ranging from 3.4 to 24.5% in red mullet fished between 1992 and 1993 in the Ligurian Sea, while in 2010–2011, red mullet has been found to be infected with *Hysterothylacium* spp. with a prevalence of up to 25.4% in the northeastern Ligurian Sea (La Spezia province). However, the fish sample size was limited to 67 individuals [20]. In other sites of the Mediterranean Sea, high prevalences of anisakid nematodes (up to 70%) have been described in fished red mullet and surmullet [21,22,23,24,25], while no *Anisakis* was found in the Ligurian sea by Klimpel et al. [26].

Given the multifaceted epidemiological situation in the Mediterranean Sea and the food hygiene problem deriving from anisakid infection in red mullet, it is of paramount importance to maintain a high level of sanitary surveillance for this widely traded fish species.

Baseline data are also needed to assess the ecological factors associated with the persistence and onset of *Anisakis* spp. in the Mediterranean Sea. To fill this gap, we investigated the prevalence of ascaridoid larvae in red mullet from two fishing sites in the Ligurian Sea.

## 2. Materials and Methods

### 2.1. Fish Sampling and Parasite Identification

This study was conducted from March 2018 to March 2020, during which a total of 838 red mullet were analyzed. Fish samples were provided by local fishers and captured between October and February through bottom trawling at two fishing sites along the coast of the Ligurian Sea (Figure 1; Imperia (N = 190) and Savona (N = 648)). Upon landing, the fish were placed in isothermal Styrofoam containers containing flake ice and transported within the same day to the Fish Diseases Laboratory at the Istituto Zooprofilattico Sperimentale del Piemonte, Liguria e Valle d’Aosta. The total body length (cm) and the body mass (g) of each fish were recorded. Following species identification, the visceral organs were removed and placed in Petri dishes containing physiological saline solution. To stimulate larvae mobilization, the Petri dishes were exposed to a heat source for at least 30 min. The musculature was cut into butterfly fillets for visual inspection. Parasite larvae detection was performed following the UV-press method described in Karl and Levsen [27]. Fillets were inspected using a UV transilluminator (UltraBright UV Transilluminator, 302 nm/365 nm, Maestrogen, Las Vegas, NV, USA) and then placed in individual plastic bags, pressed, and stored overnight at −20 °C. Genus identification was performed via light microscopy (Olympus BX40, Olympus, Tokyo, Japan) according to the morphological keys reported by Berland [28].

### 2.2. Molecular Analysis

After preliminary identification based on morphology, molecular identification of larvae was carried out using PCR-RFLP, following the protocol reported in D’Amelio et al. [29] and Pontes et al. [30].

All the *Anisakis* spp. larvae collected were analyzed using biomolecular methods. Total DNA was extracted using an ExtractMe Genomic DNA kit (Blirt S.A., Gdańsk, Poland) according to the manufacturer’s protocol for the extraction of genetic material from animal tissues. The extracted DNA underwent PCR-RFLP analysis. All reactions were performed on a 2720 Thermal Cycler (Applied Biosystems, Foster City, CA, USA) with the following master mix: 25 μL of Premix Taq ™ DNA polymerase (Takara Bio, Shiga, Japan), 0.5 μM of primers NC5 and NC2, 5 μL of extracted DNA, and ultrapure water (Sigma-Aldrich, St. Louis, MO, USA) to achieve a final volume of 50 μL. In each PCR run, DNA from a collection of *Anisakis*-positive samples (*A. pegreffii, A. simplex* s.s., *A. pegreffii* × *A. simplex* s.s. hybrid DNA) was used as positive control, while reagent blanks (ultrapure water) were included as negative controls.

The amplicons were detected via electrophoresis and underwent enzymatic cutting using HhaI, HinfI, and TaqI (Invitrogen, Carlsbad, CA, USA). The enzymatic digestion results were highlighted using 2% agarose gel electrophoresis. Fragment size was determined by comparing the results with an AmpliSize Molecular Ruler 50–2000 bp Ladder (Bio-Rad, Hercules, CA, USA).

### 2.3. Statistical Analysis

The parasitological indexes of prevalence (P), mean intensity (MI), and mean abundance (MA) of ascaridoid larvae were determined as reported in Bush et al. [31] for the two sampling sites. In addition, 95% confidence intervals (95% CIs) for the prevalence were calculated. The normality of data distribution was tested with the Shapiro–Wilk test. The biometric characteristics of the fish (total body length and body mass) are expressed as the mean ± standard deviation (SD) and displayed using a boxplot. The chi-square (χ2) test was employed to test the differences in prevalence (P) between the two sampling sites. The non-parametric Wilcoxon test was performed to determine the variation in biometric parameters between the two sampling sites. Spearman’s correlation matrix was used to determine the correlation between the number of larvae detected and biometric parameters. Finally, the non-parametric Wilcoxon test was applied to explore the difference in the number of larvae between the two sampling sites. Statistical significance was set at *p* < 0.05. Statistical analysis was performed using STATA software, version 14.2 (StataCorp., College Station, TX, USA).

## 3. Results

The average total body length of the fish was 14.04 ± 2.18 cm; the mean length of the fish from Savona was 14.35 ±2.07 cm and 12.97 ± 2.20 cm for those from Imperia. The average total body mass was 31.78 ± 17.62 g; the mean weight was 33.38 ± 16.29 g in the fish from Savona and 26.31 ± 20.67 g in the fish from Imperia (Figure 2 and Figure 3).

A total of 838 red mullet (*Mullus barbatus*) were collected from two sampling sites (Figure 1). The overall prevalence of larval stages of ascaridoid nematodes was 78.40% (657/838). The prevalence was higher in the fish from the Savona site compared to those of the Imperia site: 81.79% (530/648) versus 66.84% (127/190), respectively.

No larvae were found in the fillet musculature through UV transillumination.

Of the 657 specimens found positive for ascaridoid larvae, 4487 larvae were isolated and classified on a morphological basis, sensu Berland [28] and sensu Moravec [32], as belonging to the genus *Anisakis* (n = 544) and to the genus *Hysterothylacium* (n = 3943). Of the 544 larvae morphologically attributable to *Anisakis* spp., biomolecular analysis via PCR-RFLP identified all as *A. pegreffii*.

The overall prevalence (P) of *Anisakis* larvae was 24.46%, and the number of larvae per fish (mean intensity, MI) ranged from 1 to 11 (mean 2.65); the prevalence at the sampling site was 6.32% for Imperia (MI 1.5; range 1–6) and 29.78% for Savona (MI 2.73; range 1–11) (Table 1). There was a significant difference in the prevalence of *A. pegreffii* between the two sampling sites (χ2 = 43.79, *p* < 0.0001).

In addition, 3943 larvae morphologically identified as *Hysterothylacium* spp. were detected through visual inspection of the visceral organs (see Table 1 for prevalence, intensity, and abundance). Finally, coinfection was noted in 160 specimens: 19.09% of the total and 24.25% of the positive individuals.

Spearman’s correlation analysis showed a highly significant positive correlation between the number of *A. pegreffii* larvae and the fish body parameters (Table 2, *p* < 0.0001). There was also a significant difference between the number of *Anisakis* spp. larvae in the fish sampled at the two sampling sites (Wilcoxon test; z = 6.748; *p* < 0.05).

## 4. Discussion

Third-stage larvae of anisakid nematodes in commercial fish species constitute a serious concern for public health worldwide. Monitoring their prevalence, with particular attention to risk management in the seafood sector, is essential for consumer health. Species of the genera *Anisakis* and *Hysterothylacium* have been described in fish inhabiting the Mediterranean Sea, including, among others, *Trachurus trachurus*, *Mullus barbatus*, *Engraulis encrasicolus*, *Sardina pilchardus,* and *Merluccius merluccius* in Italy [20,33,34,35]. 

Prevalence values for *Anisakis* obtained in the present study (6.32% for Imperia and 29.78% for Savona) are consistent with those from previous epidemiological surveys on red mullet from the Ligurian Sea (range 3.4–24.5% reported in Manfredi et al. [19]). Interestingly, Klimpel et al. investigated surmullet at a different site of the Ligurian sea (Gulf of Lion) and found no *Anisakis* larvae, although only 34 fish were examined [26]. Nonetheless, previous studies in other Mediterranean areas have reported a variable, albeit generally higher, prevalence of *Anisakis* in *M. barbatus*, ranging from 2% in the Valencian coast (Spain) [21] to 41.6% in Turkey [22] and 70% in the Balearic Sea (Spain) [23]. These findings indicate a hyperendemic trend compared to our findings. Similarly, in the sibling surmullet species, analogous hyperendemic anisakid infections have been highlighted along the Libyan and Spanish Mediterranean coasts [24,25]. One explanation for the anisakid prevalence differences in Mediterranean *Mullidae* fish may be the diverse abundance of cetaceans in those areas, which are still to be explored.

No parasitic larvae were found in the musculature of the red mullet sampled, which may be explained by the short time between fishing and the laboratory examination (within the same day) and the refrigerated temperatures during the transport. Moreover, previous studies suggest a reduced ability of *A. pegreffii* to penetrate muscle tissues [35,36].

Our results show a significant positive correlation between fish body size and infection rate that can be explained by the increased parasite pressure on bigger fish, which are more likely to eat other fish as prey rather than microplankton. The varied feeding behavior of red mullet, which includes consumption of microplankton (in smaller individuals), macroplankton such as krill, and larger preys (in adult individuals), may reflect the accumulation of anisakid nematodes in adult fish, particularly those of greater length and weights. As already supported by other authors, fish size can be considered a major predictor of *Anisakis* infections [37]. Accordingly, the feeding behavior of red mullet during the spawning period may be associated with an increased probability of preying on infected intermediate hosts [38,39]. Furthermore, the accumulation of parasites in the host could also be due to the ability of larvae to survive in the host for long periods of time, even throughout the entire life span of the fish [37,40]. Nevertheless, it is important to note that since fish were regularly caught with nets compliant with fishing legislation, regulations and standards, the red mullet were found to be similar in size.

*A. pegreffii* was confirmed as the most prevalent anisakid species detected in the Ligurian Sea, as already supported by Mattiucci et al., who investigated the prevalence of anisakid nematodes in hakes from the same area [41]. The relatively high prevalence of *A. pegreffii* in the red mullet reported in our study may be explained by the inclusion of this sea in a Protected Marine Area, the Pelagos Sanctuary for Mediterranean Marine Mammals, which has the largest population of cetaceans in the western Mediterranean [42], including definitive hosts of *A. pegreffii* [41,43]. Interestingly, at the two sampling sites, only *A. pegreffii* larvae were found, which may suggest that a single *Anisakis* spp. species is circulating in the Ligurian sea, as previously reported [20,34,35]. To support this result, a denser sampling effort in the Ligurian waters is needed, including evidence of the adult nematodes in the definitive hosts. Not surprisingly, in the Mediterranean Sea, *A. pegreffii* has already been identified as the prevailing anisakid species affecting commercial fish [37,41]. 

The presence and infection dynamics of various *Anisakis* species inhabiting fishing areas may be determined using abiotic environmental factors, such as geomorphological characteristics of the sea basin, biodiversity, and abundance of marine species, as well as water temperature and salinity [44]. We found a significant difference in the prevalence of *A. pegreffii* between the two sampling sites, which are characterized by differences in the geomorphological conformation of the sea basin, possibly influencing the circulation of the marine mammals that may act as definitive hosts for *Anisakis*. Nonetheless, more information on the parasitofauna of intermediate and definitive hosts of ascaridoid nematodes in the Ligurian Sea is needed to infer the actual parasite epidemiology in this area.

The continental shelf of the Ligurian Sea is very narrow, with sea depth increasing rapidly toward the central and western parts to over 2000 m, reaching a maximum depth of 2836 m. Furthermore, the bathymetric–morphological map of the study area [45] shows numerous submarine canyons, the most notable of which are the particularly steep Genoa Canyons, located much closer to the Savona sampling site, with a maximum depth of 2000 m. The sea temperature in this area is generally constant because of canyon depth. The Imperia sampling site is further away from the canyons of Genoa; the sea is more open in this area and variable in temperature [46]. The geomorphological characteristics of the marine basin (Genoa canyons) and the environmental conditions (constant temperature) create an area of high marine biodiversity, making it an ideal habitat for phytoplankton, zooplankton, benthos, as well as cetaceans. Phytoplankton, such as the local *Posidonia* meadows, are extremely abundant in this area, creating a nursery for many species and corals. Since the temperature is constant throughout the summer months, the Genova Canyons area is extremely prolific for numerous populations of marine invertebrates and vertebrates, fish, and cetaceans that might act, respectively, as intermediate and final hosts for *Anisakis* spp. [46]. These geomorphological differences may explain the higher *Anisakis* prevalence in the Savona province, where the Genoa Canyons, which are characterized by a larger sea depth, can possibly attract cetaceans, making it a suitable habitat for them. Further epidemiological and ecological studies are required to investigate the link between cetaceans, the presence of *Anisakis* nematodes, and the role of the Genoa Canyons in facilitating this relationship. 

In conclusion, given the annual amount of red mullet captured in Italy and the anisakid prevalence reported in this study (24.46%), its persistence constitutes a serious threat for fish consumers.

What remains to be explained is the high number of parasites in the genus *Hysterothylacium* found in the fish from both sampling sites (3943 larvae of the genus *Hysterothylacium* compared to 544 of the genus *Anisakis*), which does not allow for an objective assessment of the real risk of *Anisakis.* The presence of larvae of a genus that does not pose a real health problem diminishes the quality of the fish but plays no role in causing human anisakiasis. This finding is in line with previous surveys conducted on red mullet and other fish species in the Mediterranean Sea, where high prevalences of *Hysterothylacium* spp. have been found [20,35]. Previous studies suggest that the sampling season might influence *Hysterothylacium* detection: higher prevalences are usually reported in the coldest seasons (autumn and winter) and lower ones in spring [20,25,35]. Given that the red mullet included in the present study were fished between October and February, this might explain the high prevalence obtained. We suggest an intensification of the monitoring for the presence of *Anisakis* spp. in areas with the greatest likelihood of these nematodes. Our hypothesis is that the presence of *Anisakis* spp. is influenced by optimal trophic chains, high marine biodiversity, the particular geomorphological conformation of the sea basin, and possibly by global warming and increased marine pollution.

Further studies are needed to understand the main risk factors associated with the consumption of red mullet and anisakiasis in humans along the whole food chain, from a “One health” perspective.

## 5. Conclusions

The present study provides an update on the prevalence of ascaridoid nematodes in red mullet from the Ligurian Sea. A 24.46% prevalence of *A. pegreffii* was found in red mullet, with a higher prevalence in the Imperia province. 

The results obtained can help assess the extent of the public health risk associated with *Anisakis* infection in the red mullet. Understanding the prevalence and distribution of *Anisakis* in a large sample (838 red mullet were examined in this study) can provide more accurate estimates of the potential risk to consumers. Given the implications for the fishery economy and consumer health, the prevention of zoonotic parasites in fishery products should be a priority for public health agencies and the seafood industry, as advocated in the 2010 EFSA report [47]. Further surveys with a robust sampling size are needed to understand the dynamics influencing the epidemiology of anisakid parasites in Mullidae fish in the Ligurian Sea and to better understand the ecology of this nematode in the marine ecosystem of the Mediterranean Sea. Understanding how parasites affect fish populations and the broader environment is essential for conservation efforts, especially in a protected area such as the Pelagos Sanctuary.

## Figures and Tables

**Figure 1 pathogens-12-01366-f001:**
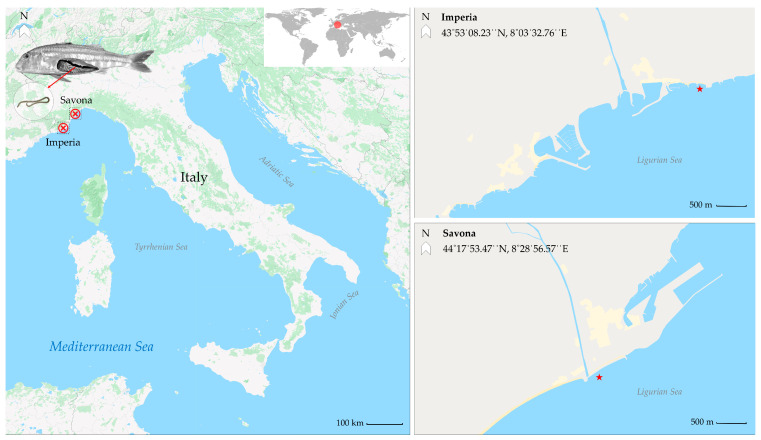
Location of the two fishing sites (Liguria, western Mediterranean Sea).

**Figure 2 pathogens-12-01366-f002:**
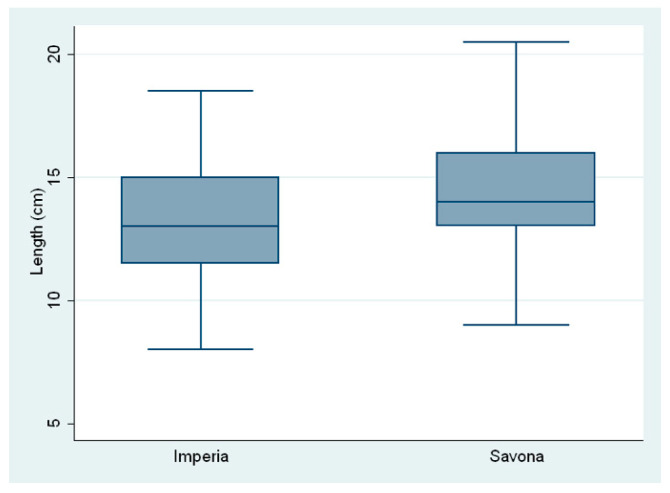
Body length of fish from the two sampling sites.

**Figure 3 pathogens-12-01366-f003:**
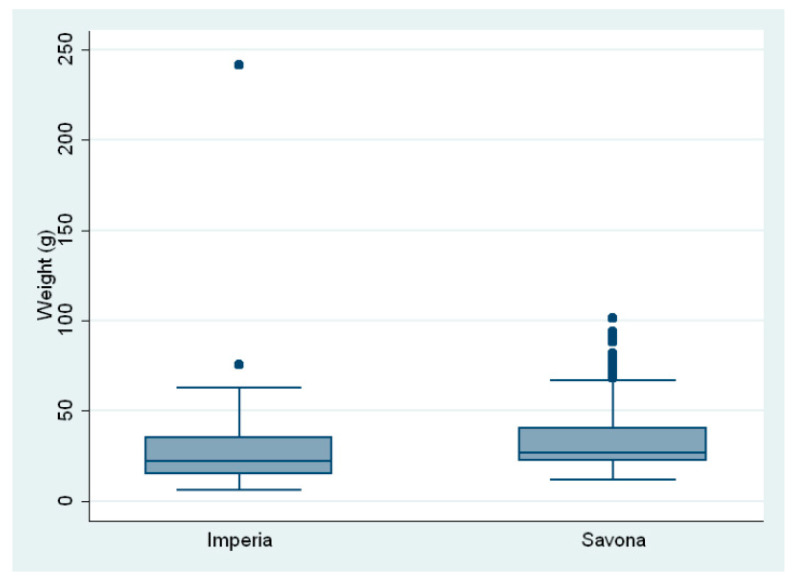
Body weight of fish from the two sampling sites.

**Table 1 pathogens-12-01366-t001:** Epidemiological indicators of *Anisakis pegreffii* and *Hysterothylacium* spp. identified in red mullet (*Mullus barbatus*) from Imperia and Savona.

Sampling Site	Red Mullet Sampled	Parasite	Infected Fish	Larvae	%P (95% CI)	MI (Range)	MA (95% CI)	Coinfected Individuals (%)
Imperia	190	Ascaridoid larvae	127	661	66.84 (60.09–73.60)	5.20 (1–130)	3.48 (2.98–4.20)	12 (6.31%)
		*Anisakis pegreffii*	12	18	6.32 (2.83–9.81)	1.5 (1–6)	0.09 (0.02–0.17)	
		*Hysterothylacium* spp.	127	643	66.84 (60.09–73.60)	5.09 (1–124)	3.38 (1.95–4.82)	
Savona	648	Ascaridoid larvae	530	3826	81.79 (78.67–84.23)	7.22 (1–70)	5.90 (5.11–6.10)	148 (22.83%)
		*A. pegreffii*	193	526	29.78 (26.25–33.31)	2.73 (1–11)	0.81 (0.68–0.94)	
		*Hysterothylacium* spp.	485	3300	74.69 (71.33–78.05)	6.80 (1–67)	5.09 (4.46–5.73)	
Total	838	Ascaridoid larvae	657	4487	78.40 (76.16–80.56)	6.82 (1–130)	5.35 (4.21–5.86)	160 (19.09%)
		*A. pegreffii*	205	544	24.46 (21.55–27.38)	2.65 (1–11)	0.65 (0.54–0.75)	
		*Hysterothylacium* spp.	612	3943	73.03 (71.98–75.15)	6.44 (1–124)	4.71 (4.12–5.29)	

**Table 2 pathogens-12-01366-t002:** Spearman’s correlation of the number of *A. pegreffii* larvae and the fish body parameters.

	Larvae (N)	Weight (g)	Length (cm)
No. of larvae	1.0000		
Weight (g)	0.427 *	1.0000	
Length (cm)	0.407 *	0.947 *	1.000

* *p* value < 0.0001.

## Data Availability

All relevant data are provided in the present study.

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
