# Peer review of "An Epidemiological Update on Anisakis Nematode Larvae in Red Mullet (Mullus barbatus) from the Ligurian Sea"

_pathogens, 2023, doi:10.3390/pathogens12111366_

Round 1

Reviewer 1 Report

Comments and Suggestions for Authors

The study aimed to monitor the presence of zoonotic nematode in one fish species intended for human consumption. The study is quite robust, however the introduction is lacking of specific information about the background, the results are not sufficiently descriptive, in particular PCR-RFLP identifications, and discussion is not focused only on results here obtained and for these reasons i suggest major revisions.

Major comments:

The introduction should provide the background that justify the study. Here, authors spend much of the introduction to describe human anisakiasis, and no other aspect of the submitted study is described. The study is about the presence of ascaridoid nematodes in red mullet, so at least they should provide already published data about Anisakis and other nematodes infecting red mullets from the Mediterranean, and also from other countries.

Regarding results, authors described the presence of A. pegreffii in all Anisakis larvae tested. They found 544 larvae morphologically attributable to Anisakis spp., but it is not clear if all 544 were processed, if all 544 gave positive PCR. Moreover, given the presence of several mammal species in the Mediterranean (and especially in the Ligurian sea) and the Anisakis eggs dispersal in the sea, it is quite uncommon that all L3 (more than 500) gave A. pegreffii profile. In fact, A. pegreffii is the prevalent species, but it is not the only species reported (for example A. simplex s.s., the hybrid, A. typica, A. physteris have been reported). I also suggest the authors to provide images of gels.

Regarding Discussion section, too many lines have been dedicated to aspects (very interesting aspects of course) that have not been included in hypothesis or methods of the study, as the climate change or pollution. Why did the authors decided to dedicate so much space to aspects that have not tested in the study? (The two study sites have not tested for temperature during the two year of sampling neither for pollution and such variables have not been included formally in a statistical comparison). I suggest to implement ecological aspect possibly related to different morphological coastal features of the two study sites and maybe use some predictive/modeling paper to speculate about aspects of interest, but not for half of the discussion (maybe only few lines).

Minor comments:

Line 51, line 60 Anisakis italics

Lines 95-96 “molecular identification of Anisakis spp. larvae was carried out by PCR-RFLP following 95 the protocol reported in Serracca et al. [32].” Please change the references about molecular methods used for identification with the two original citation included in Serracca et al, which are D’Amelio et al 2000 and Pontes et al 2005.

Lines 110-11 “All reactions are based on analysis of 110 positive (A. pegreffii, A. simplex s.s., A. pegreffii × A. simplex s.s. hybrid DNA) and negative 111 controls (ultrapure water).” Do you mean that you used positive DNA controls from the mentioned species in each PCR reaction? In the case, please rephrase. Why did you selected these species and not other? Please explain.

Line 136: change “The positive presence” with “The overall prevalence”

The footnotes in the Table 1 are not necessary. You analysed only 1 fish species and the explanation of P, MI and etc…are provided in the table legend.

Line 189: please change “our prevalence data” with “Prevalence values obtained in the present study”

Lines 197-198: “a significant positive correlation between fish body size and infection rate that might be explained by..” the positive association between fish body size and infection rate is a well know aspects of anisakis infection in fish, that tend to accumulate by time (as during the growth) larvae inside their cavities.

Author Response

REVIEWER 1

The study aimed to monitor the presence of zoonotic nematode in one fish species intended for human consumption. The study is quite robust, however the introduction is lacking of specific information about the background, the results are not sufficiently descriptive, in particular PCR-RFLP identifications, and discussion is not focused only on results here obtained and for these reasons i suggest major revisions.

Author response: Thank you for the positive feedback, we have now revised the text according to your suggestions (see below).

Major comments:

The introduction should provide the background that justify the study. Here, authors spend much of the introduction to describe human anisakiasis, and no other aspect of the submitted study is described. The study is about the presence of ascaridoid nematodes in red mullet, so at least they should provide already published data about Anisakis and other nematodes infecting red mullets from the Mediterranean, and also from other countries.

Author response: Following the Reviewer’s suggestion, we have eliminated some paragraphs on human anisakiasis in the Introduction and we have added some more information on the red mullet  epidemiological situation of anisakid infections in the Mediterranean and elsewhere. The paragraphs changed are as follows (lines 77-95)

“Manfredi et al. (2000) [19] reported a prevalence ranging from 3.4 to 24.5% in red mul-lets fished between 1992-1993 in the Ligurian Sea, while in 2010-2011, red mullet has been found infected with Hysterothylacium sp. with prevalence up to 25.4% in the north-eastern Ligurian Sea (La Spezia province), although the fish sample size was limited to 67 individuals [20].

In the Valencian coast, Anisakis simplex sensu latu (2%) and Hysterothylacium sp. (39%) were detected in 41 out of 100 (41%) red mullet in 2012 [21].

A different situation has been highlighted by Pekmezci et al. (2013) [22] and Barcala et al. (2018) [23] in Turkish costs and in the Balearic Sea (Spain) where high prevalences of anisakid nematodes up to 41% and 70% (respectively) have been described in fished red mullet.

Similarly, hyperendemic trends of anisakid larvae have been reported in the surmullet in the Libyan costs (75.2% of Anisakis sp. and 68.6% of Contracaecum sp.) [24] and in the western Mediterranean Sea (Spain) (14.29% of Anisakis and 19.05 of Hysterothylacium) [25], while no Anisakis and 8.8% of Hysterothylacium aduncum were found in the Ligu-rian sea by Klimpel et al. (2008) [26].

Given the multifaceted epidemiological situation in the Mediterranean Sea and the food hygiene problem deriving from anisakid infection in the red mullet, it is of para-mount importance to maintain high the sanitary surveillance of this widely traded fish species.”   

Regarding results, authors described the presence of A. pegreffii in all Anisakis larvae tested. They found 544 larvae morphologically attributable to Anisakis spp., but it is not clear if all 544 were processed, if all 544 gave positive PCR. Moreover, given the presence of several mammal species in the Mediterranean (and especially in the Ligurian sea) and the Anisakis eggs dispersal in the sea, it is quite uncommon that all L3 (more than 500) gave A. pegreffii profile. In fact, A. pegreffii is the prevalent species, but it is not the only species reported (for example A. simplex s.s., the hybrid, A. typica, A. physteris have been reported). I also suggest the authors to provide images of gels.

Author response: All the collected larvae were first morphologically identified and then submitted to molecular analysis (only the end tail portion of the worm). Since this was a very large sample, the molecular ID was not performed all in one session, and unfortunately we have not taken the images of gels, also because nowadays is something that is not asked anymore to show in the publications and not mandatory for the journal’s guidelines.
Regarding the explanation why only A. pegreffii was found, we think that this is not uncommon in our sampling area, as other authors reported the same finding in the past, confirming that Anisakis pegreffii is the most prevalent anisakid species detected in Ligurian Sea (see: Serracca et al., 2013; Menconi et al., 2021; Mattiucci et al. 2004).

We have modified the text (line 134) as follows:
“All the Anisakis spp. larvae collected were analysed…”

Regarding Discussion section, too many lines have been dedicated to aspects (very interesting aspects of course) that have not been included in hypothesis or methods of the study, as the climate change or pollution. Why did the authors decided to dedicate so much space to aspects that have not tested in the study? (The two study sites have not tested for temperature during the two year of sampling neither for pollution and such variables have not been included formally in a statistical comparison). I suggest to implement ecological aspect possibly related to different morphological coastal features of the two study sites and maybe use some predictive/modeling paper to speculate about aspects of interest, but not for half of the discussion (maybe only few lines).

Author response: Thank you for the suggestion. Indeed, we had wrongly included in the aim of the study the possible association between geomorphology and parasite onset, but we didn’t actually test this hypothesis, so we decided to delete this aim of the study.
We have also modified the Discussion section adding some more paragaphs on the epidemiological aspects (see lines 347-353)

Minor comments:

Line 51, line 60 Anisakis italics

Author response: Done

Lines 95-96 “molecular identification of Anisakis spp. larvae was carried out by PCR-RFLP following 95 the protocol reported in Serracca et al. [32].” Please change the references about molecular methods used for identification with the two original citation included in Serracca et al, which are D’Amelio et al 2000 and Pontes et al 2005.

 Author response: the two references have been added in the text.

Lines 110-11 “All reactions are based on analysis of 110 positive (A. pegreffii, A. simplex s.s., A. pegreffii × A. simplex s.s. hybrid DNA) and negative 111 controls (ultrapure water).” Do you mean that you used positive DNA controls from the mentioned species in each PCR reaction? In the case, please rephrase. Why did you selected these species and not other? Please explain.

Author response: According to the Reviewer’s comment, we modified the sentence as follows (lines 142-144) : “In each PCR run, DNA from a collection of Anisakis-positive samples (A. pegreffii, A. simplex s.s., A. pegreffii × A. simplex s.s. hybrid DNA) was used as positive control, while reagent blanks (ultrapure water) was included as negative controls.”
We decided to select these Anisakis species because these are the ones that have been described in the Ligurian sea according to the scientific literature (see: Serracca et al., 2013; Menconi et al., 2021; Mattiucci et al. 2004).

Line 136: change “The positive presence” with “The overall prevalence”

Author response: Ok, done.

The footnotes in the Table 1 are not necessary. You analysed only 1 fish species and the explanation of P, MI and etc…are provided in the table legend.

Author response: Ok, footnotes have been deleted.

Line 189: please change “our prevalence data” with “Prevalence values obtained in the present study”

Author response: Ok, done.

Lines 197-198: “a significant positive correlation between fish body size and infection rate that might be explained by..” the positive association between fish body size and infection rate is a well know aspects of anisakis infection in fish, that tend to accumulate by time (as during the growth) larvae inside their cavities.

Author response: We completely agree with the Reviewer, and this is why we provided a reference supporting this aspect which was already explored. We added a sentence as follows: “

“As already supported by other authors, fish size can be considered a major predictor of Anisakis infections [40].”

Reviewer 2 Report

Comments and Suggestions for Authors

It is an interesting paper becuse is specific enought to claim attention for tha practical problem of anisakiasis and anisakidiosis on an important consuminf fish as is Mullus barbatus. The paper is easy to read and the methodology is comprehenssible. My recomendation is to accept after minor changes as are:

a) Figure 1 seems to have poor deffinition.

b) Paragraph 2.2. Molecular Analysis

In the text it is not written the used method (D´Amelio, Mattiucci, etc). It has to be included in the reference list.

c) Table 2.

Remove the p values which are under the correlation values (it is explainning in the text refering to table 2 in line 176 ".....and the fish body parameters (Table 2, p<0.0001)."

d) Discussion

Line 198: introduce between "....might be explained by" and "the varied feeding.....".a text such as "the need of food under parasitic demands."

Line 200: Introduce after "[40]" the text ", favour such a hypothetical assumtion"

The paragraph would be: "....might be explained by the need of food under parasitic demands. The varied feeding behaviour of red mullet, which eat  microplankton (smaller individuals), macroplankton as krill, or larger prey as adult fish [40], favour such a hypothetical assumtion."

Author Response

REVIEWER 2

It is an interesting paper becuse is specific enought to claim attention for tha practical problem of anisakiasis and anisakidiosis on an important consuminf fish as is Mullus barbatus. The paper is easy to read and the methodology is comprehenssible. My recomendation is to accept after minor changes as are:

  1. a) Figure 1 seems to have poor deffinition.

Author response: The Figure has been changed and now it has a higher definition.

  1. b) Paragraph 2.2. Molecular Analysis

In the text it is not written the used method (D´Amelio, Mattiucci, etc). It has to be included in the reference list.

Author response: the two references have been added in the text.

  1. c) Table 2.

Remove the p values which are under the correlation values (it is explainning in the text refering to table 2 in line 176 ".....and the fish body parameters (Table 2, p<0.0001)."

Author response: the “zeros” of the p values have been removed from table 2. We have added in the Table caption the explanation of the p value.

  1. d) Discussion

Line 198: introduce between "....might be explained by" and "the varied feeding.....".a text such as "the need of food under parasitic demands."

Author response: Ok, done.

Line 200: Introduce after "[40]" the text ", favour such a hypothetical assumtion"

Author response: A sentence has been added following the other Reviewers’ suggestions: ““As already supported by other authors, fish size can be considered a major predictor of Anisakis infections [40].”

The paragraph would be: "....might be explained by the need of food under parasitic demands. The varied feeding behaviour of red mullet, which eat  microplankton (smaller individuals), macroplankton as krill, or larger prey as adult fish [40], favour such a hypothetical assumtion."

Author response: Thank you for the suggestion, we modified the text accordingly:
 “The varied feeding behaviour of red mullet, which eat microplankton (smaller individuals), macroplankton as krill, or larger prey as adult fish, favours such a hypo-thetical assumption. As already supported by other authors, fish size can be considered a major predictor of Anisakis infections .”

Reviewer 3 Report

Comments and Suggestions for Authors

From my point of view, this is a well written manuscript covering a topic of interest regarding the parasitological infection of the M. barbatus in the Mediterranean area. However, I believe it needs some changes in order to be published.

I think the manuscript is inconsistent in the topics included in the introduction in comparison with those in the discussion. Anisakiosis symptomatology is too widely explained in the introduction while in the discussion some biotic and abiotic aspects are extensively discussed when nothing is mentioned in the introduction.

Moreover, in the last paragraph of the introduction, when talking about the objectives of the work, authors say “we tested the hypothesis that Anisakis life cycle and prevalence may be influenced by geomorphological features, water temperature, and ichthyofauna biodiversity of the Ligurian Sea”. The factors influencing the life cycle and prevalence are very wide and can be extensively discussed. In your discussion, sometimes you use the M. barbatus example and sometimes you talk in general. From my point of view, the two topics, the epidemiological study of M. barbatus and how this prevalence is influenced, should be more related. It is true that authors discuss their results in regards the geographical characteristics but I think the discussion should be extended including other geographical areas.

I would also like to mention that Hysterothylacium species are not included in the Anisakidae family, they are in the Raphidascarididae.

Finally, from my point of view the conclusions do not reflect the main topics of the manuscript, only the first sentence.   

Title: Personally, I do not like the second part of the title “and a plausible hypothesis for the onset of parasitosis”. Firstly, including the word “plausible” is a little bit pretentious from my point of view. I would delete it. Regarding the rest of the sentence and after reading the discussion, I understand why authors have chosen this title, but an hypothesis for the parasitosis is a wide concept. It does not convince me.

Keywords: I would delete “food hygiene”. Do you mention it anywhere along then manuscript? “geomorphological features of sea basin is very long. I would try to find another word. Finally, zoonotic parasites is redundant if you already included Anisakis pegreffii in the keywords. In general, for the last five keywords I would try to find fewer words, which included those you have decided.

Introduction:

As explained before, I believe you talk too much about the symptomatology and the illness when afterwards anything is discussed. I would reduce it a lot. I also think something should be said about Hysterothylacium, as it is the most detected species in your work. I think you should also say something about the A. simplex (s.l.) sibling species complex, as A. pegreffii is included in this complex.

Lines 32-33: From my point of view the first sentence does not link with the rest of the paragraph, where you only talk about the host. I would move it to another part of the introduction.

Lines 33 and 35: You start two consecutive sentences with “Red mullet”. Please rephrase one of them.

Lines 38 to 40: By how it is written, it is not clear if the 6500 tones are regarding the annual catch worldwide or in Italy? The link regarding this reference does not work.

Lines 41 to 66: I would change the order of this paragraphs as you talk first about the disease, then about the life cycle and finally about the symptomatology. Maybe it would be better if you talk about the life cycle and then about the illness/symptomatology.

Lines 42 and 43: I would remove “and cause of economic losses worldwide” and move it to somewhere else in the introduction because in this paragraph you are talking about the disease.

Line 44: A point is missing after the last s of “(s.s)”. It should be “(s.s.)”.

Line 51: Anisakis should be in italics. Please check all the manuscript as afterwards (for example in line 60) occurs the same.

Lines 53 to 66: As said before, I think this paragraph I too long.

Lines 64 to 66: I think this last sentence does not link with the rest of the paragraph.

Lines 67 to 72: From my point of view this paragraph is not clear enough. You say you want to fill a gap but it is regarding the epidemiological data in Italy or for assessing the ecological factors? For both objectives, studying the red mullet in the Ligurian Sea is enough? I would rewrite it.

Materials and methods

Line 76: In the section title, I would add something regarding the parasites, as you talk more than only fish sampling in this section.

Line 77 and line 80: The difference between number of hosts analysed from the two sampling sites is significant. Why? Were they sampled at the same period of the year?

Lines 94 to 96: This is molecular identification, should be moved to the molecular section. Also Serraca et al. is not the correct reference. The first authors performing the PCR-RFLP with the different restrictive enzymes were Zhu et al, 1998 and D’Amelio et al. 2000.

Lines 102 to 115: Please add the reference.

Lines 109 to 112: I would delete these lines, specially the second one. From my point of view, they are not necessary.

Results

You talk about Anisakidae, mixing Anisakis with Hysterothylacium, and this is wrong, as Hysterothylacium belong to the Raphidascarididae family.

In materials and methods, you say you performed the UV methodology for the visualization of larvae in the flesh but you do not have any results here. Where there no larvae in the fish’s musculature? It could be useful afterwards for discussing the human risk of red mullet.

Line 136: I would change “positive presence” for prevalence.

Lines 136 to 138: Is this difference statistically significant? If it is please say it here and include p<0.05.

Line 141: In material and methods authors said that the identification was carried out until genus level, however, I think if you add another reference, you could say you found A. simplex (s.l.), especially if afterwards all Anisakis larvae are identified as A. pegreffii.

Lines 142 and 143: Please, delete “according to the protocol reported in Serraca”. It is unnecessary as you already explained in the material and methods section.

Lines 148 and 149: The visual inspection method has also been explained in material and methods. You could mix the two first sentences.

Lines 160 and 161: I do not understand why this is mentioned here. Assuming would not be discussing?

Lines 161 to 163: This should be moved to the previous part of the results.

Lines 163 to 166: I think the fish characteristics should be moved to the beginning of the results, as it is the first thing you do when fish arrive to the laboratory and they are the first results you obtain.  

Line 177 and 178: In this sentence, are you referring only to A. pegreffii or to both Anisakis and Hysterothylacium larvae?

Discussion:

As said before, the topic you are discussing, the influence on the parasite prevalence, is extensive and a widely discussion could be made. From my point of view your discussion should be more linked to your host, especially if you want to compare two different areas. In some parts, you link it properly (for example regarding the geomorphological area) but in other parts, I think you do not. It is fine if you hypothesize in general, but I think you should do it once you have done it comparing you samples.

Also, when hypothesizing, I understand you use your results but I think you should include more in your discussion other works regarding parasitation of the red mullet, in the Mediterranean Seaand also in other areas.

When comparing the two sampling sites, there is a significant difference regarding Anisakis parasitation but something should be said about the number of hosts studied as 190 vs 648 is a considerable difference.

As you said in the objectives, you want to discuss about the life cycle of anisakis but you do not mention anything about the definitive hosts. Is not there a protected cetacean area in the Ligurian Sea? Could it have any influence in the prevalence of anisakis?  

Finally, I think you write two many small paragraphs. Sometimes a single sentence is written as a paragraph. You should try to assemble the sentences into paragraphs as much as possible.

Lines 190 to 193: Here there is a thing I do not understand. You say your results are consistent with the findings of these two works of the Ligurian Sea, understanding the Ligurian Sea as an “homogenic” area. However, afterwards, you focus a lot in the difference of your two sampling sites. Where did the other works sample they fish? Are they similar to one of your two sampling sites?

Lines 192 to 196: In the same line as the previous comment, you compare it with other regions of the Mediterranean. Are those regions that different in comparison with you sampling sites? I think in the posterior discussion, regarding aspects influencing parasitation, this works should also be discussed.

Lines 197 to 199: A single sentence for a paragraph? Please link it to the next paragraph in this case. The same in lines 206 to 207 and other parts of the discussion.

Lines 208 to 218: In the introduction, you say the red mullet is usually found between 10 and 300. Regarding this, do you think this submarine canyons influence in the red mullet life? How?

Lines 211 and 212: Could this significant difference be due to the number of hosts studied? I think something should be mentioned.

Lines 253 to 255: I do not understand what are you summarizing here. Also you mix the amount of red mullet captured in Italy with you results of parasitation in you study which is only of the Ligurian Sea and then you talk about risk for consumers.

Lines 256 to 261: What do other authors say about Hysterothylacium parasitation in red mullet of the Ligurian Sea? And in other areas of Italy/Europe?

Conclusions

Line 274: You say two sampling sites in the Mediterranean but due to the proximity of the two sampling sites, I think instead of Mediterranean, you should talk about the Ligurian Sea.

Lines 274 to 279: I do not really see how are these sentences are conclusions of your work.

References:

Please check all your references again. The doi format is not always the same (for example line 300 and line 316). Journal name should be in italics and the year of publication in bold (for example line 300 is wrong)

Table: The title of the table is simply the titles of the columns. Please try to find a title which includes all the concepts without repeating all column tittles. If you include a column of infected fish I think the column of non infected is not necessary. You could talk, in the table title or in the column as M. barbatus instead of fish so you would not need the 1. Also I think you should try to change the N, as you use the same letter for different things (fish and larvae). As mentioned before, Anisakidae should not include Hysterothylacium.

Author Response

REVIEWER 3

From my point of view, this is a well written manuscript covering a topic of interest regarding the parasitological infection of the M. barbatus in the Mediterranean area. However, I believe it needs some changes in order to be published.

I think the manuscript is inconsistent in the topics included in the introduction in comparison with those in the discussion. Anisakiosis symptomatology is too widely explained in the introduction while in the discussion some biotic and abiotic aspects are extensively discussed when nothing is mentioned in the introduction.

Moreover, in the last paragraph of the introduction, when talking about the objectives of the work, authors say “we tested the hypothesis that Anisakis life cycle and prevalence may be influenced by geomorphological features, water temperature, and ichthyofauna biodiversity of the Ligurian Sea”. The factors influencing the life cycle and prevalence are very wide and can be extensively discussed. In your discussion, sometimes you use the M. barbatus example and sometimes you talk in general. From my point of view, the two topics, the epidemiological study of M. barbatus and how this prevalence is influenced, should be more related. It is true that authors discuss their results in regards the geographical characteristics but I think the discussion should be extended including other geographical areas.

I would also like to mention that Hysterothylacium species are not included in the Anisakidae family, they are in the Raphidascarididae.

Finally, from my point of view the conclusions do not reflect the main topics of the manuscript, only the first sentence.  

 Author response: Thank you for your constructive feedback, we have now remodulated the Introduction and Discussion sections according to your and other Reviewer’s suggestions. See responses below for a more detailed description of the modifications.

We have laso changed the term Anisakidae with ascaridoid nematodes or larvae in the text.

Title: Personally, I do not like the second part of the title “and a plausible hypothesis for the onset of parasitosis”. Firstly, including the word “plausible” is a little bit pretentious from my point of view. I would delete it. Regarding the rest of the sentence and after reading the discussion, I understand why authors have chosen this title, but an hypothesis for the parasitosis is a wide concept. It does not convince me.

Author response: We decided to delete the second part of the title, which is now “An epidemiological survey of Anisakis nematode larvae in red mullet (Mullus barbatus) from the Ligurian Sea”.

Keywords: I would delete “food hygiene”. Do you mention it anywhere along then manuscript? “geomorphological features of sea basin is very long. I would try to find another word. Finally, zoonotic parasites is redundant if you already included Anisakis pegreffii in the keywords. In general, for the last five keywords I would try to find fewer words, which included those you have decided.

Author response: food hygiene has been deleted, geomorphological features of sea basin has been changed with “Mediterranean sea”, we changed fish-borne zoonotic parasites with fish-borne parasite and zoonoses.

Introduction:

As explained before, I believe you talk too much about the symptomatology and the illness when afterwards anything is discussed. I would reduce it a lot. I also think something should be said about Hysterothylacium, as it is the most detected species in your work. I think you should also say something about the A. simplex (s.l.) sibling species complex, as A. pegreffii is included in this complex.

Author response: Following the Reviewer’s 1 and your suggestion, we have eliminated some paragraphs on human anisakiasis in the Introduction and we have added some more information on the red mullet  epidemiological situation of anisakid infections in the Mediterranean and elsewhere. The paragraphs changed are as follows in lines 77-95)

“Manfredi et al. (2000) [19] reported a prevalence ranging from 3.4 to 24.5% in red mul-lets fished between 1992-1993 in the Ligurian Sea, while in 2010-2011, red mullet has been found infected with Hysterothylacium sp. with prevalence up to 25.4% in the north-eastern Ligurian Sea (La Spezia province), although the fish sample size was limited to 67 individuals [20].

In the Valencian coast, Anisakis simplex sensu latu (2%) and Hysterothylacium sp. (39%) were detected in 41 out of 100 (41%) red mullet in 2012 [21].

A different situation has been highlighted by Pekmezci et al. (2013) [22] and Barcala et al. (2018) [23] in Turkish costs and in the Balearic Sea (Spain) where high prevalences of anisakid nematodes up to 41% and 70% (respectively) have been described in fished red mullet.

Similarly, hyperendemic trends of anisakid larvae have been reported in the surmullet in the Libyan costs (75.2% of Anisakis sp. and 68.6% of Contracaecum sp.) [24] and in the western Mediterranean Sea (Spain) (14.29% of Anisakis and 19.05 of Hysterothylacium) [25], while no Anisakis and 8.8% of Hysterothylacium aduncum were found in the Ligu-rian sea by Klimpel et al. (2008) [26].

Given the multifaceted epidemiological situation in the Mediterranean Sea and the food hygiene problem deriving from anisakid infection in the red mullet, it is of para-mount importance to maintain high the sanitary surveillance of this widely traded fish species.”   

Lines 32-33: From my point of view the first sentence does not link with the rest of the paragraph, where you only talk about the host. I would move it to another part of the introduction.

Author response: We agree. The sentence has been moved to line 46.

Lines 33 and 35: You start two consecutive sentences with “Red mullet”. Please rephrase one of them.

Author response: We have moved the sentence below and now the word “red mullet” is not repeated.

Lines 38 to 40: By how it is written, it is not clear if the 6500 tones are regarding the annual catch worldwide or in Italy? The link regarding this reference does not work.

Author response: The sentence was referred to annual catch in Italy. We modified accordingly.

Lines 41 to 66: I would change the order of this paragraphs as you talk first about the disease, then about the life cycle and finally about the symptomatology. Maybe it would be better if you talk about the life cycle and then about the illness/symptomatology.

Author response: we have now reformulated all the paragraph eliminating the parts regarding symptomatology. We think that the order of the sentence is now more comprehensible.

Lines 42 and 43: I would remove “and cause of economic losses worldwide” and move it to somewhere else in the introduction because in this paragraph you are talking about the disease.

Author response: We think that in the context of the study the term “economic loss” is pertinent, as anisakiasis implies costs for public health too, considering hospitalized people, laboratory costs etc.

Line 44: A point is missing after the last s of “(s.s)”. It should be “(s.s.)”.

Author response: Added as suggested.

Line 51: Anisakis should be in italics. Please check all the manuscript as afterwards (for example in line 60) occurs the same.

Author response: Modified accordingly.

Lines 53 to 66: As said before, I think this paragraph I too long.

Author response: we have now reformulated all the paragraph eliminating the parts regarding symptomatology (see introduction section lines 77-90).

Lines 64 to 66: I think this last sentence does not link with the rest of the paragraph.

Author response: This sentence has been eliminated.

Lines 67 to 72: From my point of view this paragraph is not clear enough. You say you want to fill a gap but it is regarding the epidemiological data in Italy or for assessing the ecological factors? For both objectives, studying the red mullet in the Ligurian Sea is enough? I would rewrite it.

Author response: The paragraph has been rephrased (see lines 77-100) as follows:
“Given the multifaceted epidemiological situation in the Mediterranean Sea and the food hygiene problem deriving from anisakid infection in the red mullet, it is of paramount importance to maintain high the sanitary surveillance of this widely traded fish species.  

Baseline data are also needed to assess the ecological factors associated with the persistence and onset of Anisakis spp in the Mediterranean Sea. To fill this gap, we investigated the prevalence of ascaridoid larvae in red mullet from two fishing sites in the Ligurian Sea.”

Materials and methods

Line 76: In the section title, I would add something regarding the parasites, as you talk more than only fish sampling in this section.

Author response: the heading has been modified in “Fish Sampling and parasite identification”

Line 77 and line 80: The difference between number of hosts analysed from the two sampling sites is significant. Why? Were they sampled at the same period of the year?

Author response: The number of fish sampled per site was opportunistic (based on the possibility of the selected fisherman) so this is why the number of fish was different in the two sites. We only set a minimum number of samples per site, which was 100. The sampling has been carried out in the same period of the year for the two sampling sites (October- February). (added in line 107).

Lines 94 to 96: This is molecular identification, should be moved to the molecular section. Also Serraca et al. is not the correct reference. The first authors performing the PCR-RFLP with the different restrictive enzymes were Zhu et al, 1998 and D’Amelio et al. 2000.

Author response: We moved this sentence to the following paragraph and we modified the references.

Lines 102 to 115: Please add the reference.

Author response: The references have been already added in the first paragraph, all the molecular paragraph is the brief explanation of the procedure.

Lines 109 to 112: I would delete these lines, specially the second one. From my point of view, they are not necessary.

Author response: We believe that describing the positive and negative controls in the PCR reaction might be an important information for the reader, therefore we decided to keep this the sentence. We deleted the other sentence on the thermal protocols.

Results

You talk about Anisakidae, mixing Anisakis with Hysterothylacium, and this is wrong, as Hysterothylacium belong to the Raphidascarididae family.

Author response: we have now changed with “ascaridoid nematodes”.

In materials and methods, you say you performed the UV methodology for the visualization of larvae in the flesh but you do not have any results here. Where there no larvae in the fish’s musculature? It could be useful afterwards for discussing the human risk of red mullet.

Author response: We are sorry about this mistake. Indeed, we did not find any anisakid  larvae in the musculature, but this is most likely due to the fact that the fish were examined shortly after the collection in the sea. Therefore, the larvae did not have the time to migrate postmortem. We now added a sentence in the Result section (line 188): “No larvae were found in the fillet musculature through UV transillumination.”

Line 136: I would change “positive presence” for prevalence.

Author response: Modified as suggested.

Lines 136 to 138: Is this difference statistically significant? If it is please say it here and include p<0.05.

Author response: The difference was tested and we stated the significant p value. This sentence has now been moved in line 197-8

Line 141: In material and methods authors said that the identification was carried out until genus level, however, I think if you add another reference, you could say you found A. simplex (s.l.), especially if afterwards all Anisakis larvae are identified as A. pegreffii.

Author response: Yes, indeed the morphological identification was carried out until genus level and then, by molecular means, until species level.

Lines 142 and 143: Please, delete “according to the protocol reported in Serraca”. It is unnecessary as you already explained in the material and methods section.

Author response: Ok, done.

Lines 148 and 149: The visual inspection method has also been explained in material and methods. You could mix the two first sentences.

Author response: we changed the sentence as follows: “In addition, 3943 larvae morphologically identified as Hysterothylacium sp. were detected by visual inspection in the visceral organs (see Table 1 for prevalence, intensity, and abundance).”

Lines 160 and 161: I do not understand why this is mentioned here. Assuming would not be discussing?

Author response: We agree with the Reviewer, we moved this sentence in the Discussion section (lines 232-233)

Lines 161 to 163: This should be moved to the previous part of the results.

Author response: This sentence has been moved to the previous part of the results.

Lines 163 to 166: I think the fish characteristics should be moved to the beginning of the results, as it is the first thing you do when fish arrive to the laboratory and they are the first results you obtain. 

Author response: Ok, results on fish biometry have been moved at the beginning of the results section.

Line 177 and 178: In this sentence, are you referring only to A. pegreffii or to both Anisakis and Hysterothylacium larvae?

Author response: We are referring to Anisakis spp.

Discussion:

As said before, the topic you are discussing, the influence on the parasite prevalence, is extensive and a widely discussion could be made. From my point of view your discussion should be more linked to your host, especially if you want to compare two different areas. In some parts, you link it properly (for example regarding the geomorphological area) but in other parts, I think you do not. It is fine if you hypothesize in general, but I think you should do it once you have done it comparing you samples.

Also, when hypothesizing, I understand you use your results but I think you should include more in your discussion other works regarding parasitation of the red mullet, in the Mediterranean Seaand also in other areas.

When comparing the two sampling sites, there is a significant difference regarding Anisakis parasitation but something should be said about the number of hosts studied as 190 vs 648 is a considerable difference.

As you said in the objectives, you want to discuss about the life cycle of anisakis but you do not mention anything about the definitive hosts. Is not there a protected cetacean area in the Ligurian Sea? Could it have any influence in the prevalence of anisakis?

Author response: We agree with the Reviewer that Discussion section had to be improve, so we added some more information regarding parasitation of surmullet in the Mediterranean Sea and other areas.
Regarding definitive hosts, unfortunately we do not have information on cetaceans gastrointestinal parasites in the Ligurian Sea, as this is area is included in the Protected Marine Area, the Pelagos Sanctuary for Mediterranean Marine Mammals, which actually has the largest population of cetaceans in the Western Mediterranean (Notarbartolo di Sciara et al., 2008), including definitive hosts of A. pegreffii.  We added this information in the Discussion as well.

Finally, I think you write two many small paragraphs. Sometimes a single sentence is written as a paragraph. You should try to assemble the sentences into paragraphs as much as possible.

Author response: We modified the Discussion section following the Reviewer’s suggestion. There are now less small paragraphs.

Lines 190 to 193: Here there is a thing I do not understand. You say your results are consistent with the findings of these two works of the Ligurian Sea, understanding the Ligurian Sea as an “homogenic” area. However, afterwards, you focus a lot in the difference of your two sampling sites. Where did the other works sample they fish? Are they similar to one of your two sampling sites?

Author response:

Lines 192 to 196: In the same line as the previous comment, you compare it with other regions of the Mediterranean. Are those regions that different in comparison with you sampling sites? I think in the posterior discussion, regarding aspects influencing parasitation, this works should also be discussed.

Author response: The areas compared with our sampling sites (Mediterranea Sea, Balearic cost, Lybian costs) are probably different in terms of geomorphology and biodiversity. Nonetheless, we think that speculations on these aspect would be out of the scope of our study.

Lines 197 to 199: A single sentence for a paragraph? Please link it to the next paragraph in this case. The same in lines 206 to 207 and other parts of the discussion.

Author response: We have implemented this paragraph as follows: “Our results show a significant positive correlation between fish body size and infection rate that can be explained by the increased need of food under higher parasitic demands. Nevertheless, it is important to note that, since fish were regularly caught with nets compliant with fishing legislation, regulations, and standards, the red mullets were found to be similar in size.”

Lines 208 to 218: In the introduction, you say the red mullet is usually found between 10 and 300. Regarding this, do you think this submarine canyons influence in the red mullet life? How?

Author response: We think that submarine canyons can influence red mullet life, thus Anisakis ecology too somehow, as fish living in more shallow water (eg where submarine canyons exist) will less likely to be in the trophic food chain of Anisakis (no predators and less preys).
Unfortunately we are not aware of population dynamics studies on red mullet in the Ligurian Sea, and therefore we cannot draw valid conclusions on this point.

Lines 211 and 212: Could this significant difference be due to the number of hosts studied? I think something should be mentioned.

Author response: No we don’t think that the significant difference is related to the different sampling number, as in both sampling sites the sample size was quite robust.

Lines 253 to 255: I do not understand what are you summarizing here. Also you mix the amount of red mullet captured in Italy with you results of parasitation in you study which is only of the Ligurian Sea and then you talk about risk for consumers.

Author response: We changed “summarizing” with “in conclusion”. The sentence is now as follows:
“In conclusion, given the annual amount of red mullet captured in Italy and Anisakid prevalence reported in this study (24.46%), its persistence constitutes a serious and permanent risk of human anisakiasis for consumers.”

Lines 256 to 261: What do other authors say about Hysterothylacium parasitation in red mullet of the Ligurian Sea? And in other areas of Italy/Europe?

Author response: We added in line 337-9 the following paragraph: “This finding is in line with previous surveys in the red mullet and other fish species in the Mediterranean Sea, where high prevalences of Hysterothylacium have been found (Serracca et al., 2013; Menconi et al., 2022).”

Conclusions

Line 274: You say two sampling sites in the Mediterranean but due to the proximity of the two sampling sites, I think instead of Mediterranean, you should talk about the Ligurian Sea.

Author response: Modified as suggested.

Lines 274 to 279: I do not really see how are these sentences are conclusions of your work.

Author response: We have now implemented the conclusions with the following sentence:
“Further surveys with robust sampling size are needed to explore the epidemiological situation of anisakid parasites in Mullidae fish in the Ligurian Sea, and to better understand the ecology of this nematode within and outside the Mediterranean Sea.”

References:

Please check all your references again. The doi format is not always the same (for example line 300 and line 316). Journal name should be in italics and the year of publication in bold (for example line 300 is wrong)

Author response: the references have been revised and they should be now correct.

Table: The title of the table is simply the titles of the columns. Please try to find a title which includes all the concepts without repeating all column tittles. If you include a column of infected fish I think the column of non infected is not necessary. You could talk, in the table title or in the column as M. barbatus instead of fish so you would not need the 1. Also I think you should try to change the N, as you use the same letter for different things (fish and larvae). As mentioned before, Anisakidae should not include Hysterothylacium.

Author response: We modified the Table heading, eliminated the non-infected column and the abbreviations.

Round 2

Reviewer 1 Report

Comments and Suggestions for Authors

The manuscript is improved, however i still find statements in the discussion that are not related to the present investigation. 

For example:

Lines 295-296 and following "Moreover, increased pollution is an additional factor in the prevalence of Anisakis parasites." --> if you want to discuss this aspect, that is not in the aim of your study, you should consider to mitigate at least the verb using might instead of is, or to say another concomitant potential aspect that migth have an effect on the prevalence....

anyway from 281 to 303 i suggest to remove, it is out of scope to discuss something that is not carried out in the study

Author Response

REVIEWER 1 

The manuscript is improved, however i still find statements in the discussion that are not related to the present investigation.  

For example: 

Lines 295-296 and following "Moreover, increased pollution is an additional factor in the prevalence of Anisakis parasites." --> if you want to discuss this aspect, that is not in the aim of your study, you should consider to mitigate at least the verb using might instead of is, or to say another concomitant potential aspect that migth have an effect on the prevalence.... 

 Author response: we have changed the verb as suggested. “Moreover, increased pollution might be an additional factor in the prevalence” 

anyway from 281 to 303 i suggest to remove, it is out of scope to discuss something that is not carried out in the study 

Author response: we have eliminated the text between lines 281-303. 

Reviewer 3 Report

Comments and Suggestions for Authors

I think the manuscript has improved but I think it still needs to be modified in order to be published. Some of the questions asked in the first review were not answered or addressed. Please see my suggestions below.

Introduction

Lines 31 to 38: As the paper is focused on the Mullus barbatus, I would not talk about the Mullus surmulentus in this paragraph. I would rewrite it all only mentioning the Mullus barbatus. I think the way it is written confuses the reader. For example in line 33 you name the Mullus barbatus as red mullet while in the previous sentence you named the mullus surmuletus also as red mullet.

Line 47: You say “A. pegreffii in particular”, but this is not true, as worldwide both species have been the cause of the disease with no significant difference between the two sibling species. Moreover, the reference you use for this particularity does not mention anything about A pegreffii being the primary cause of anisakiasis.

Line 50: The life cycle can also involve only two hosts, when infested crustaceans are eaten by cetaceans.

Lines 60 and 61: I think this sentence should not be a paragraph.

Lines 62 to 77: I think the way this information is written is like in a discussion. I would rewrite it in a more general way. Also, this information is repeated, as it is discussed afterwards in the discussion section.

Material and Methods

Lines 100 to 106: I think the UV methodology should not be as deeply explained because afterwards you do not find any larvae, meaning you did not see any fluorescent larvae, did you?

Line 114: I think you should eliminate Pontes et al and Serraca et al, or at least one the two references.

Line 134: In the way it is written (“as reported in”), I think you should write the author of the reference.

Line 167: I would include this sentence to a paragraph.

Discussion

Lines 212 to 219: I asked for this aspect in the first review but I did not get any answer. You say your results are consistent with the findings of these two works of the Ligurian Sea, understanding the Ligurian Sea as an “homogenic” area. However, afterwards, you focus a lot in the difference of your two sampling sites. Where did the other works sample they fish? Are they similar to one of your two sampling sites?

Lines 220 and 221: This sentence is on paragraph.

Lines 224 to 225: I think you could use this sentence in another part of the discussion. I think that, specifically in this paragraph is contradictory, as the Ligurian Sea is a protected area for cetaceans so the no parasitation of the surmullet is rare.

Line 232: Is there any reference to support your hypothesis regarding the increased need of food under high parasitic demands? Do you consider your values of prevalence and abundance as high parasitic demand? I think something regarding the relation between length and age of the fish should be said.

Line 237: Which hypothetical assumption?

Lines 245 and 246: I think this sentence is not pertinent here. I would move it to another paragraph.

Line 247: This sentence is wrong because you found more Hysterothylacium than A. pegreffii. It should be rewritten in regards to the intention of the authors.

Lines 249 to 251: I do not understand what you are talking about when you say “These findings”. You mean the elevated abundance described by Mattiucci? Do you think your prevalence and abundance values could be also considered high? Should not be all fish species of the Ligurian see heavily parasitized if they are in a protected area for cetaceans?

Lines 283 to 303: I think this part of the discussion is unnecessary. You are discussing global warming in general. I do not see any relation with your findings regarding the red mullet. Personally I would delete it, if you are very reluctant to do it, I would suggest to reduce it.

Conclusions

Lines 331 to 336: I also mentioned this in my previous review. I do not really see how are these sentences are conclusions of your work. I see them as discussion.

References:

There are still some mistakes, for example Doi in line 407 or the year without bold letters in line 480. Please ensure all the bibliography is properly written.

References 1 and 2 start in a different line compared to the rest.

Table 1

I still believe the title of the table is too repetitive as it is very similar with the first row. Could you find a sentence resuming all the concepts instead of repeating them?

I would delete the column of infected fish, as you already include the number of fish sampled and the prevalence.

Maybe, in the same column of the coinfection I would include the percentage between brackets.

MI and MA are not %. Please delete it.

Author Response

REVIEWER 3 

 I think the manuscript has improved but I think it still needs to be modified in order to be published. Some of the questions asked in the first review were not answered or addressed. Please see my suggestions below. 

Introduction 

Lines 31 to 38: As the paper is focused on the Mullus barbatus, I would not talk about the Mullus surmulentus in this paragraph. I would rewrite it all only mentioning the Mullus barbatus. I think the way it is written confuses the reader. For example in line 33 you name the Mullus barbatus as red mullet while in the previous sentence you named the mullus surmuletus also as red mullet. 

Author response: We have corrected the name of surmullet as “striped mullet or surmullet” and now there should be no confusion on the terminology between the two mullet species. We think that it is important to mention both species in the first part of the introduction, as they are taxonomically correlated and (as other reviewers asked) we put them in both introduction and discussion section to compare our results with other studies obtained in surmullet and red mullet in different areas. 

Line 47: You say “A. pegreffii in particular”, but this is not true, as worldwide both species have been the cause of the disease with no significant difference between the two sibling species. Moreover, the reference you use for this particularity does not mention anything about A pegreffii being the primary cause of anisakiasis. 

Author response: We agree with you, we have now eliminated “in particular” (line 47). 

Line 50: The life cycle can also involve only two hosts, when infested crustaceans are eaten by cetaceans. 

Author response: we completely agree with you, we have now adjusted the paragraph this way: 
“The life cycle of anisakid nematodes involves at least two different hosts, including marine mammals as final hosts where adult nematodes develop in the stomach, crustaceans (i.e., euphasiids) as intermediate hosts where free-swimming L3 larvae are ingested and develop in the hemocoel, and many different fish families and cephalopods as intermediate or paratenic hosts [11-12].” 

Lines 60 and 61: I think this sentence should not be a paragraph. 

Author response: We have now connected the two paragraphs. 

Lines 62 to 77: I think the way this information is written is like in a discussion. I would rewrite it in a more general way. Also, this information is repeated, as it is discussed afterwards in the discussion section. 

Author response: We have now shortened the paragraph (lines 62-69) accordingly, as follows: Manfredi et al. [19] reported a prevalence ranging from 3.4 to 24.5% in red mullets fished between 1992-1993 in the Ligurian Sea, while in 2010-2011, red mullet has been found infected with Hysterothylacium sp. with prevalence up to 25.4% in the north-eastern Ligurian Sea (La Spezia province), although the fish sample size was limited to 67 individuals [20]. 
In the other sites of the Mediterranean Sea, high prevalences of anisakid nematodes (up to 70%) have been described in fished red mullets and surmullets [21-25], while no Anisakis was found in the Ligurian sea by Klimpel et al. [26]. 

Material and Methods 

Lines 100 to 106: I think the UV methodology should not be as deeply explained because afterwards you do not find any larvae, meaning you did not see any fluorescent larvae, did you? 

Author response: We did not find any larvae in the fillet, correct. Nonetheless, we think that the UV methodology should be explained regardless of the results obtained (in our case, negative results). 

Line 114: I think you should eliminate Pontes et al and Serraca et al, or at least one the two references. 

Author response: we have eliminated Serracca et al. 

Line 134: In the way it is written (“as reported in”), I think you should write the author of the reference. 

Author response: we have added the name of the author (Bush et al). 

Line 167: I would include this sentence to a paragraph. 

Author response: Done 

Discussion 

Lines 212 to 219: I asked for this aspect in the first review but I did not get any answer. You say your results are consistent with the findings of these two works of the Ligurian Sea, understanding the Ligurian Sea as an “homogenic” area. However, afterwards, you focus a lot in the difference of your two sampling sites. Where did the other works sample they fish? Are they similar to one of your two sampling sites? 

Author response: The study by Manfredi et al (2000) is the one we cited to compare our results, as they sampled red mullet from the Ligurian Sea. According to the georeferencing in the methods described by Manfredi et al., the exact location of the sampling site is Imperia (44°N and 8°E), although they did not specify it in the text. 

Our speculations on the geomorphology are based on the fact that Genova Canyons are located much closer to the Savona sampling site than the Imperia site. 

This means that in the Savona area sea is much deeper (it can reach depth of 2836 m) than the Imperia site, favouring the free circulation of large marine mammals wich may act as definitive host for Anisakis spp. This might explain the higher prevalence of the Savona province compared to Imperia, but it does not explain why Manfredi et al (2000) found a higher prevalence of Anisakis compared to ours in the same area (3.4-24.5% compared to 6.32%). 

We have added in line 258 this sentence “....sea basin, possibly influencing the circulation of marine mammals that may act as definitive hosts for Anisakis” 

Moreover, we have added some paragraphs to clarify our speculations in line 280-7: “These geomorphological differences may explain the higher Anisakis prevalence in the Savona province, where the Genoa Canyons, which are characterized by a higher sea depth, can possibly attract cetaceans, making it a suitable habitat for them. 

Further epidemiological and ecological studies would be required to investigate the link between cetaceans, the presence of Anisakis nematodes, and the role of the Genoa Canyons in facilitating this relationship”. 

Lines 220 and 221: This sentence is on paragraph. 

Author response: It has been merged with the previous paragraph. 

Lines 224 to 225: I think you could use this sentence in another part of the discussion. I think that, specifically in this paragraph is contradictory, as the Ligurian Sea is a protected area for cetaceans so the no parasitation of the surmullet is rare. 

Author response: It’s true that the findings of Klimpel is in contrast with our results and in general, with what is expected to be found, but it has to be reminded that the sample size was very limited (only 34 surmullets were examined, meaning almost 1/25 of our sample size), thus the fish sample might have been non-representative. 

Line 232: Is there any reference to support your hypothesis regarding the increased need of food under high parasitic demands? Do you consider your values of prevalence and abundance as high parasitic demand? I think something regarding the relation between length and age of the fish should be said. 

Author response: The sentence was not correct as it was, so we modified it in line 229: “Our results show a significant positive correlation between fish body size and infection rate that can be explained by the increased parasite pressure of bigger fish which are more likely to eat fish as preys” 

The reference that we provided (Levsen et al., 2018) supported the idea that larger body fish size is associated to higher parasitic loads, which can be explained by the switch of food behaviour from microplankton to larger preys which may be parasitized. We modified the sentence for more clarity: “the varied feeding behaviour ….may reflect the cumulation of anisakid nematodes in adult fish (thus, of greater length and weights). 

Regarding prevalence and abundance in our study, we consider the overall Anisakis spp prevalence obtained (24.46%) normal from an ecological point of view (considering that marine mammals circulates in the area), but relatively high from a sanitary point of view. It depends on which perspective you are looking at the data.  

Line 237: Which hypothetical assumption? 

Author response: We have reformulated the sentence as follows: “the varied feeding behaviour ….may reflect the cumulation of anisakid nematodes in adult fish (thus, of greater length and weights).” 

Lines 245 and 246: I think this sentence is not pertinent here. I would move it to another paragraph. 

Author response: We have now eliminated this sentence. 

Line 247: This sentence is wrong because you found more Hysterothylacium than A. pegreffii. It should be rewritten in regards to the intention of the authors. 

Author response: We have adjusted the sentence accordingly. “A. pegreffii is confirmed as the most prevalent anisakid species detected in the Ligurian Sea, as already supported by Mattiucci et al. in other fish species in the same area” 

Lines 249 to 251: I do not understand what you are talking about when you say “These findings”. You mean the elevated abundance described by Mattiucci? Do you think your prevalence and abundance values could be also considered high? Should not be all fish species of the Ligurian see heavily parasitized if they are in a protected area for cetaceans? 

Author response: We referred to our findings (which are supported by other studies in the same area). We consider 24.46 a “relatively high” prevalence of Anisakis in the red mullet, considering that the sample size is consistent (838 red mullet sampled), thus most probably representative. We are not aware of studies that have investigated the same number of red mullet (e.g., in the study by Mattiucci et al., (2004), they have investigated 150 hakes from the Ligurian Sea), and this could have influenced the prevalence obtained. 
We cannot infer on the prevalence of definitive hosts because we don’t have recent updated data on gastrointestinal parasites in cetaceans in the Ligurian sea. Mattiucci reported that “The highest abundances recorded for this species (A. Pegreffi) were in samples from the Ligurian Sea; perhaps because this area includes the ‘Sanctuary for Cetaceans’, a Mediterranean area inhabitated by various dolphin species, such as the bottlenose dolphin Tursiops truncatus (Montagu), one of the main definitive hosts of A. pegreffii (Nascetti et al., 1986; Mattiucci et al., 1997; unpubl. data).” 

Since red mullet are considered good preys for cetaceans such as the bottlenose dolphin , we assume that the prevalence of adult anisakid in cetaceans in the Ligurian Sea would be quite high, but this is only an assumption. On the contrary, red mullet infection with Anisakis as intermediate or paratenic host strictly depends on the diet of this fish species, which is variable depending on the age and on the depth of sea inhabited. 

This is why we stated that “more information on the parasitofauna of intermediate and definitive hosts of ascaridoid nematodes in the Ligurian Sea are needed to infer the actual epidemiology in this area.” 

Lines 283 to 303: I think this part of the discussion is unnecessary. You are discussing global warming in general. I do not see any relation with your findings regarding the red mullet. Personally I would delete it, if you are very reluctant to do it, I would suggest to reduce it. 

Author response: As also suggested by Reviewer 1, we have eliminated the text between lines 288-303. 

Conclusions 

Lines 331 to 336: I also mentioned this in my previous review. I do not really see how are these sentences are conclusions of your work. I see them as discussion. 

Author response: Conclusions should summarize the key findings of a study, and their implications, highlighting the significance of the study and providing recommendations for future research/next steps. We have now implemented the conclusions of this study as follows (divided per section): 
- significance of the study and summary of results: “The present study provides an update on the prevalence of ascaridoid nematodes in red mullet from two sampling sites in the Ligurian Sea. A 24.46% prevalence of A. pegreffii third stage larvae was found in the red mullet, with a higher prevalence in the Imperia province. 
The results obtained can help assess the extent of the public health risk associated with Anisakis infection in red mullet. Understanding the prevalence and distribution of Anisakis in a large sample (838 red mullet examined in this study) can provide more accurate estimates of the potential risk to consumers.” 
- implications: “Given the implications for fishery economy and consumer health, prevention of zoonotic parasites in fishery products should be a priority for public health agencies and the seafood industry. Over the last decades, information on Anisakid nematodes has improved, and the European Food Safety Authority (EFSA) strongly recommends continuing research, as advocated in the 2010 EFSA report [62].”  

  • Suggestions for future research: Further surveys with robust sampling size are needed to understand the dynamics influencing the epidemiology of anisakid parasites in Mullidae fish in the Ligurian Sea, and to better understand the ecology of this nematode in the marine ecosystem of the Mediterranean Sea. 
    Understanding how parasites affect fish populations and the broader environment is essential for conservation efforts, especially in a protected area such as the Pelagos Sanctuary.” 

References: 

There are still some mistakes, for example Doi in line 407 or the year without bold letters in line 480. Please ensure all the bibliography is properly written. 

References 1 and 2 start in a different line compared to the rest. 

Author response: We have now checked and corrected the bibliography accordingly. 

Table 1 

I still believe the title of the table is too repetitive as it is very similar with the first row. Could you find a sentence resuming all the concepts instead of repeating them? 

I would delete the column of infected fish, as you already include the number of fish sampled and the prevalence. 

Maybe, in the same column of the coinfection I would include the percentage between brackets. 

MI and MA are not %. Please delete it. 

Author response: According to the Reviewer suggestion, we have changed the title of the table as follows: “Prevalence (P), mean intensity (MI), and mean abundance (MA) of Anisakis pegreffii and Hysterothylacium spp. identified in red mullet (Mullus barbatus) from Imperia and Savona.” 
We think that the number of infected fish can be a useful information for a reader to have in a separate column, so we prefer to keep it. We have deleted % in MI and MA and we have included % in the column of the coinfection in brackets. 

Round 3

Reviewer 3 Report

Comments and Suggestions for Authors

I believe the manuscript has improved but from my point of view some minor changes should be done in order to be published. Before following with my comments, I would like to mention that some changes already commented in the previous reviews have not been addressed. I believe this is a lack of professionalism, specially regarding the bibliography, as I still have found some mistakes.

Lines 31 to 32: I already mentioned this in the previous review.  Along all the manuscript you refer to the Mullus barbatus as red mullet, but here the red mullet, between brackets, is referred to Mullus surmuletus. Why? Please, change it. 

Lines 94 to 97: Although I do not share your opinion, I could understand that you want to mention the UV transillumination method. However, I strongly believe that these two specific sentences should be deleted as you did not see the frozen nematode larvae nor its brightly fluorescence. Also, you did not isolate, rinse in Petri dishes nor count them, as you did not find any larvae in the flesh. If the second sentence “Genus identification…. Berland”, is not referring exclusively to larvae of the flesh you should keep it, but if it refers only to larvae of the flesh you should also delete it. 

Lines 201 and 202: The way it is written it seems that the only fish species from Italy parasitized with Anisakis and Hysterothylacium are those five species mentioned in the text. I would include “among others” at some point of the sentence.

Lines 212 to 213: I would move this sentence to line 205, when you are talking to the epidemiological results of the Ligurian Sea.

Lines 214 to 216: This is a one sentence paragraph. I mentioned this many times in the last revisions.

Line 217: This paragraph does not start with the correct indent. Please change it here and along the manuscript as it also occurs many times, for example in lines 221, 244, 247 etc.

Lines 234 to 236: This sentence is repeated.

Line 244 to 246: This is a one sentence paragraph. I would join it to the next paragraph.

Lines 254 to 257: I would move this paragraph to line 243, where you talk about A. pegreffii. Also, regarding the A. pegreffii discussion in the Ligurian sea, I think you should mention something about the Mediterranean Sea. A. pegreffi is by far the most abundant species in the Mediterranean, so to find it in the Ligurian is not an extraordinary finding.

Lines 278 to 283: Again, here there are two paragraphs with only one sentence. I understand that sometimes a single sentence might be a paragraph but I believe it is not the case of any of the times you have done it in your manuscript.

Lines 306 to 323: I know what conclusions should summarize but the sentence in lines 315 to 317 is without any doubt a discussion sentence. I strongly believe that it has nothing to do with the conclusions of your work. Also, I would not divide the conclusions in three different paragraphs. I would leave it in a single paragraph.

Lines 341 to 460: Since the first revision one of my comments was regarding the references and there are still mistakes. I think this is a strong lack of professionalism.

The two first references present a different indent.

Please, confirm that refence number 7 is well cited. If it is a webpage should not be the last day of entrance cited?

In reference 11, are you sure the ISBN is mandatory?

Some “doi” are underlined and some others are not. Which is the correct way?

Some “doi” are also missing, for example reference 17 has a DOI. Please check all your references properly.

The year in Reference 35 is not in bald.

Table 1: In the last revision I suggested to find a sentence which included all the parasitological indexes but they are still repeated in the title and in the table.  

Author Response

REVIEWER 3

I believe the manuscript has improved but from my point of view some minor changes should be done in order to be published. Before following with my comments, I would like to mention that some changes already commented in the previous reviews have not been addressed. I believe this is a lack of professionalism, specially regarding the bibliography, as I still have found some mistakes.

 Author response: We thank the Reviewer for his/her time. We apologize if some mistakes in the bibliography were still present.

Lines 31 to 32: I already mentioned this in the previous review.  Along all the manuscript you refer to the Mullus barbatus as red mullet, but here the red mullet, between brackets, is referred to Mullus surmuletus. Why? Please, change it.

Author response: The names were inverted by mistake, apologies. The common names in brackets have now been corrected.

Lines 94 to 97: Although I do not share your opinion, I could understand that you want to mention the UV transillumination method. However, I strongly believe that these two specific sentences should be deleted as you did not see the frozen nematode larvae nor its brightly fluorescence. Also, you did not isolate, rinse in Petri dishes nor count them, as you did not find any larvae in the flesh. If the second sentence “Genus identification…. Berland”, is not referring exclusively to larvae of the flesh you should keep it, but if it refers only to larvae of the flesh you should also delete it.

Author response: We are sorry about this, the sentence was erroneously kept in the methods although we did not find any larvae. We eliminated the sentence regarding the frozen larvae, while we kept the sentence on the genus identification.

Lines 201 and 202: The way it is written it seems that the only fish species from Italy parasitized with Anisakis and Hysterothylacium are those five species mentioned in the text. I would include “among others” at some point of the sentence.

Author response: Ok, we have added “among others” in the sentence in line 201, as suggested.

Lines 212 to 213: I would move this sentence to line 205, when you are talking to the epidemiological results of the Ligurian Sea.

Author response: The sentence has been moved in line 206 as suggested.

Lines 214 to 216: This is a one sentence paragraph. I mentioned this many times in the last revisions.

Author response:

We have rephrased this sentence as follows: “One explanation for the Anisakid prevalence diversity differences in Mullidae fish from different areas of the Mediterranean Sea may be the diverse abundance of cetaceans in those areas, which is still to be explored.” and merged the sentence with previous paragraph, in order not to appear “one sentence paragraph”.
We’re not completely sure that this is what the Reviewer meant by his/her comment (and the previous in the last revision). With track change revisions it is very difficult to control paragraph differentiation as they might appear differently with track changes.

Line 217: This paragraph does not start with the correct indent. Please change it here and along the manuscript as it also occurs many times, for example in lines 221, 244, 247 etc.

Author response: We have corrected the indents as suggested. As previously mentioned, the many changes along the text in track change have made the indent view editing more difficult.

Lines 234 to 236: This sentence is repeated.

Author response: This is odd because in the revised manuscript sent for the last revision (Docx version titled “ pathogens-264063”) it does not appear repeated, while in the PDF generated by the system it appears repeated. Anyway, we have checked and there are no repeated statements now.

Line 244 to 246: This is a one sentence paragraph. I would join it to the next paragraph.

Author response: The sentence has been moved in line 254 as suggested.

Lines 254 to 257: I would move this paragraph to line 243, where you talk about A. pegreffii. Also, regarding the A. pegreffii discussion in the Ligurian sea, I think you should mention something about the Mediterranean Sea. A. pegreffi is by far the most abundant species in the Mediterranean, so to find it in the Ligurian is not an extraordinary finding.

Author response: The paragraph has been moved as suggested. A sentence on the Mediterranean was also added in lines 249-251: “Not surprisingly, in the Mediterranean Sea, A. pegreffii has already been identified as the prevailing anisakid species affecting commercial fish [37, 41].

Lines 278 to 283: Again, here there are two paragraphs with only one sentence. I understand that sometimes a single sentence might be a paragraph but I believe it is not the case of any of the times you have done it in your manuscript.

Author response: We have corrected the paragraph.

Lines 306 to 323: I know what conclusions should summarize but the sentence in lines 315 to 317 is without any doubt a discussion sentence. I strongly believe that it has nothing to do with the conclusions of your work. Also, I would not divide the conclusions in three different paragraphs. I would leave it in a single paragraph.

Author response: The paragraphs are We have now reduced paragraphs and eliminated the sentence highlighted by the reviewer, merging it with the previous one. “Given the implications for fishery economy and consumer health, prevention of zoonotic parasites in fishery products should be a priority for public health agencies and the seafood industry, as advocated in the 2010 EFSA report.”

Lines 341 to 460: Since the first revision one of my comments was regarding the references and there are still mistakes. I think this is a strong lack of professionalism.

The two first references present a different indent.

Author response: Ok, changed.

Please, confirm that refence number 7 is well cited. If it is a webpage should not be the last day of entrance cited?

Author response: The accession date has been added in REF N.7

In reference 11, are you sure the ISBN is mandatory?

Author response: we have eliminated the ISBN

Some “doi” are underlined and some others are not. Which is the correct way?

Author response: Some of them were underlined (or in blue) because the default settings create an hyperlink when adding the DOI in the reference list. Anyway, the DOI will be formatted by the journal to create a link so we don’t think this is a big deal.

Some “doi” are also missing, for example reference 17 has a DOI. Please check all your references properly.

Author response: DOI has been added in ref 17 and the remaining references, when available.

The year in Reference 35 is not in bald.

Author response: Ok corrected.

Table 1: In the last revision I suggested to find a sentence which included all the parasitological indexes but they are still repeated in the title and in the table. 

Author response: We are not sure what the Reviewer is asking. We propose this different table caption:  “Epidemiological indicators of Anisakis pegreffii and Hysterothylacium spp. identified in red mullet (Mullus barbatus) from Imperia and Savona.”. Although all the paper we cited use the same “formula” to indicate table captions in the prevalence table.